# Look on Demand: A Cognitive Scheduling Framework for Visual Evidence Acquisition in Multimodal Reasoning

Yang Zhang [1]  Xiaoshuai Sun [1 2]  Rui Zhao [3]  Wujin Sun [4]  Yidong Chen [3 4]  Jiayi Ji [1]  Qian Chen [5]  Rongrong Ji [1]

## Abstract

Existing multimodal reasoning approaches mainly follow two paradigms: pre-reasoning visual-to-text conversion or reasoning in a unified vision–language space. The former compresses fine-grained visual details through static textualization, while the latter suffers from linguistic dominance, weakening faithfulness to visual evidence. In this work, we argue that a central challenge is how and when visual evidence is introduced into the reasoning process. Motivated by this insight, we propose a multimodal reasoning framework in which a language model maintains a reasoning state and dynamically schedules visual evidence acquisition, deciding both when to query an independent perception module and when to terminate reasoning. Experiments across multiple multimodal reasoning benchmarks show that CSMR consistently outperforms representative baselines in accuracy under zero-shot settings. Further analysis demonstrates that these gains stem from reasoning-state-driven visual querying and early termination. The code is available at https://github.com/YangZhang2511/CSMR

## 1. Introduction

In recent years, the rapid advancement of large-scale Vision–Language Models (VLMs) has significantly accelerated research on multimodal reasoning. A central goal of this line of work is to enable models to perform complex reasoning by effectively integrating visual and linguistic infor-

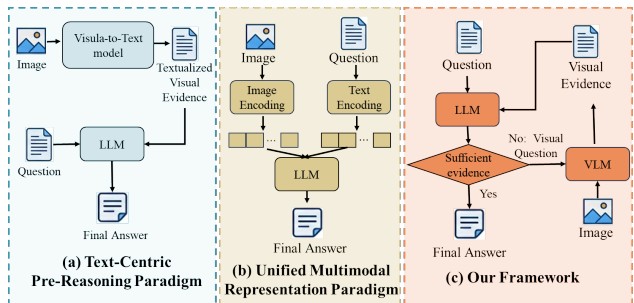

*Figure 1.* Illustration of two dominant multimodal reasoning paradigms and our framework.

mation. Depending on how visual evidence is introduced during reasoning, existing approaches can be broadly categorized into two paradigms. The first paradigm converts visual inputs into textualized visual evidence before reasoning, and subsequently performs the entire reasoning process purely in the language space (Yang et al., 2022; Salaberria et al., 2023; Zheng et al., 2023), as illustrated in Fig. 1(a). This design directly leverages the strong reasoning capabilities of large language models (LLMs). However, since the reasoning trajectory has not yet unfolded at the time of one-shot visual-to-text conversion, the initial textualization can only capture coarse semantics, inevitably compressing or discarding fine-grained evidence that may become decisive at later stages (Zhang et al., 2024a; Bigverdi et al., 2025). The second paradigm introduces visual evidence directly into the reasoning process at the representation level (Mitra et al., 2024; Li & Ma, 2025; Lei et al., 2025; Man et al., 2025), enabling end-to-end reasoning within a unified vision–language embedding space, as illustrated in Fig.1(b). By avoiding explicit visual-to-text conversion, this paradigm retains direct access to raw visual representations, thereby preventing the irreversible loss of fine-grained visual evidence, and has become a prevalent approach in recent multimodal reasoning research. However, as discussed in Section 4, this paradigm exhibits a structural bias that causes visual representations to be influenced by linguistic priors, resulting in weakened grounding in image evidence.

Taken together, these observations suggest that the two paradigms correspond to two typical failure modes: either

---

[1]Key Laboratory of Multimedia Trusted Perception and Efficient Computing, Ministry of Education of China, Xiamen University, 361005, P.R. China. [2]Sino-Russian Research Center for Digital Economy. [3]School of Informatics, Xiamen University, China. [4]Institute of Artificial Intelligence, Xiamen University, China. [5]School of Information Engineering, Xiamen Ocean Vocational College, Xiamen 361102, China. Correspondence to: Xiaoshuai Sun <xssun@xmu.edu.cn>.

*Proceedings of the 43rd International Conference on Machine Learning*, Seoul, South Korea. PMLR 306, 2026. Copyright 2026 by the author(s).

important visual evidence is lost during the conversion of visual inputs into text, or visual representations are undermined in their faithfulness to image evidence. These observations indicate that the central challenge in multimodal reasoning lies in determining when and how visual evidence should be introduced and integrated into the reasoning process. Inspired by the working memory theory of Baddeley et al. (Baddeley, 2020), we propose CSMR, a cognitive scheduling framework for visual evidence acquisition in multimodal reasoning. As illustrated in Fig. 1(c), an LLM serves as the cognitive core that controls the reasoning process and dynamically invokes a VLM during inference to obtain necessary visual evidence. Because the acquisition of visual evidence is explicitly driven by the reasoning state, it is no longer treated as a one-time conversion, but rather as a process that can be repeatedly invoked and progressively validated. Even if the visual evidence returned by the perception module is insufficient, the reasoning core can continue to invoke the perception module until sufficient visual evidence is obtained to support the reasoning process. Meanwhile, the structural decoupling of perception and reasoning prevents the extracted visual evidence from being influenced by linguistic priors. Our contributions are threefold:

1.We analyze a structural limitation in unified multimodal reasoning: visual representations are influenced by linguistic priors, resulting in weakened grounding in image evidence.

2.We propose CSMR, in which a language model maintains an explicit reasoning state and uses it to govern the reasoning process, dynamically invoking an independent visual perception module to acquire task-relevant visual evidence.

3.We conduct extensive evaluations on multiple multimodal reasoning benchmarks, demonstrating consistent improvements in accuracy over representative baselines.

## 2. Related Work

### 2.1. Text-Centric Pre-Reasoning Paradigm

In this paradigm, visual inputs are converted into textual representations prior to reasoning, and all subsequent reasoning is performed purely in the language space without further access to the original images. Early approaches rely on global image descriptions for vision-to-text conversion (Yang et al., 2022; Salaberria et al., 2023), while later works improve task alignment by introducing question-relevant captions (Hu et al., 2023) or more abstract intermediate textual representations, such as candidate-answer-based reasoning (Shao et al., 2023). DDCoT (Zheng et al., 2023) further decomposes problems into sub-questions, queries visual evidence via external tools, and integrates the resulting textual evidence for final reasoning. Despite their effectiveness, these

approaches rely on static and pre-planned textualization of visual inputs, preventing the model from incrementally supplementing or revising visual evidence during reasoning, which limits performance on tasks requiring fine-grained or iterative visual verification.

### 2.2. Unified Multimodal Representation Paradigm

This paradigm performs end-to-end reasoning by fusing visual and textual features into a unified multimodal representation space. By jointly modeling both modalities, these methods aim to enhance cross-modal interaction during reasoning. Existing approaches can be roughly grouped into three categories. Early methods, such as T5 (Raffel et al., 2020), fuse visual and textual features but generate text-only reasoning outputs. Subsequent works introduce image-related intermediate structures or prompting signals to guide reasoning in the shared space, including CCoT (Mitra et al., 2024) and SCAFFOLD (Lei et al., 2025). More recent methods further intensify multimodal interaction by jointly generating visual and textual information during reasoning, enabling multimodal reasoning chains, as exemplified by ICoT (Gao et al., 2025) and AIMCoT (Li & Ma, 2025). Despite these advances, this paradigm is prone to hallucinations (Deng et al., 2024; Liu et al., 2025b; Wang et al., 2025; Yang et al., 2025); this paper further analyzes the underlying causes of this issue. As analyzed in Section 4, under joint multimodal training objectives, dominant linguistic priors may bias visual representations toward the language space, undermining faithfulness to the original image evidence.

In addition, some interactive or tool-augmented reasoning approaches (Chen et al., 2023; Yang et al., 2023; Gupta & Kembhavi, 2023) mainly reason by imitating demonstration trajectories, making them essentially trajectory imitation-driven. Thus, both query generation and answer prediction tend to follow demonstrated patterns rather than being guided by the internal reasoning state of the current task.

## 3. Preliminaries

In this paper, we focus on the *single-stream* paradigm, which is the dominant architecture in contemporary VLMs such as LLaVA (Li et al., 2024) and Qwen-VL (Team, 2025). A typical single-stream VLM consists of a vision encoder $E_v$ and a decoder-only LLM, which process visual and textual information within a unified token sequence. Given an image $I$ and a textual question $q$, the vision encoder encodes the image into visual embeddings:

$$x^{\text{vis}} = E_v(I). \tag{1}$$

The question is converted into an instruction-style prompt $\text{Prompt}_{\text{instruct}}$ and encoded into textual embeddings:

$$x^{\text{text}} = E_t(\text{Prompt}_{\text{instruct}}(q)).\qquad(2)$$

where $E_t(\cdot)$ denotes the text embedding function of the language model, which maps the input text into a sequence of embeddings. The resulting visual and textual embeddings are concatenated to form a joint multimodal representation:

$$X = [x^{\text{vis}}; x^{\text{text}}].\qquad(3)$$

Finally, $L$ autoregressively generates the answer sequence:

$$p_\Theta(a_t \mid X, a_{<t}) = \text{LLM}(X, a_{<t}), \qquad t = 1, \ldots, T,\quad(4)$$

producing the final output

$$a_{\text{final}} = (a_1, a_2, \ldots, a_T).\qquad(5)$$

During answer generation, visual and textual information is integrated solely through the LLM's self-attention mechanism. Formally, the attention weight assigned to the $j$-th token at layer $\ell$ is

$$A_{ij}^{(\ell)} = \frac{\exp\left(s_{ij}^{(\ell)}\right)}{\sum\limits_{k\in[X;\,a_{<i}]} \exp\left(s_{ik}^{(\ell)}\right)},\qquad(6)$$

where the attention score is computed by

$$s_{ij}^{(\ell)} = \frac{Q_i K_j^\top}{\sqrt{d}},\qquad(7)$$

with $Q_i$ denoting the query vector at the $i$-th decoding position, $K_j$ the key vector of the $j$-th token, and $d$ the dimensionality of the query and key vectors.

# 4. Empirical Analysis

This section analyzes why the unified multimodal representation paradigm systematically undermines faithful visual grounding in the vision encoder $E_v$. We show that this issue stems from two structural factors: (i) the training objective lacks explicit constraints on visual faithfulness, and (ii) the language-prior-dominated self-attention mechanism biases the optimization of the vision encoder.

## 4.1. Limitation of the Standard Training Objective

The standard optimization objective does not enforce that the vision encoder $E_v$ produce representations faithfully grounded in the image. Formally, given an input image $I$ and a question $q$, the training objective is

$$\Theta^* = \arg\max_\Theta p_\Theta(a_{\text{final}} \mid I, q),\qquad(8)$$

where $\Theta = \{\theta_v, \theta_l\}$ denote the parameters of the $E_v$ and $L$. Under this objective, the model may achieve a low training loss without fully relying on visual evidence. This is because the LLM $L$ can leverage strong linguistic priors to produce outputs that align with the target answer $a_{\text{final}}$. To verify this point, we conducted an experiment in the Appendix A. The results show that the model retains surprisingly high accuracy even without images, indicating a strong reliance on linguistic priors.

## 4.2. Language-Dominant Attention Causes Misleading Visual Updates

The vision encoder is prone to misleading updates during training because visual tokens have limited influence on the final prediction. This weakness arises from two attention-related issues: (i) text tokens consistently gain higher attention weights than visual tokens, and (ii) the reasoning paradigm produces long textual chains that dilute the already limited attention assigned to visual tokens.

**Intrinsic Attention Bias Toward Text Tokens.** During answer generation, the model concentrates most attention on text tokens while assigning substantially less to visual tokens. This systematic bias originates from the internal attention mechanism of the LLM $L$. As described in Section 3, when generating the next token $a_{i+1}$, the model allocates attention over the entire sequence $[X; a_{<i+1}]$. The attention assigned to visual tokens is:

$$A_{\text{vis}}^{(\ell)} = \frac{\sum\limits_{j\in x^{\text{vis}}} \exp\left(s_{ij}^{(\ell)}\right)}{\sum\limits_{j\in x^{\text{vis}}} \exp\left(s_{ij}^{(\ell)}\right) + \sum\limits_{k\in[x^{\text{text}};\,a_{<i+1}]} \exp\left(s_{ik}^{(\ell)}\right)}.\quad(9)$$

Visual tokens typically receive lower attention scores for two reasons: (1) the query token $a_i$ originates from the text domain, and its query vector $Q_i$ is optimized during pretraining to better match text tokens; (2) the representation space is shaped by large-scale text corpora, making textual tokens densely structured while visual tokens are comparatively sparse and semantically weaker in this space, which systematically reduces $Q_i$–$K_j$ similarity.

To substantiate this analysis, we analyze the attention distribution between visual and textual tokens when generating the first answer token in Qwen3-VL-8B on a subset of ScienceQA (Appendix A.2). As shown in Fig. 2, across all 35 Transformer layers, the average pre-softmax attention score on text tokens is consistently higher than that on image tokens (about $2.5\times$ on average). Softmax normalization further magnifies this logit gap, systematically reducing the attention mass assigned to visual tokens (Chen et al., 2024a; Zhang et al., 2025; Zhu et al., 2026). Our findings align with prior work (Deng et al., 2025), which shows that vision-

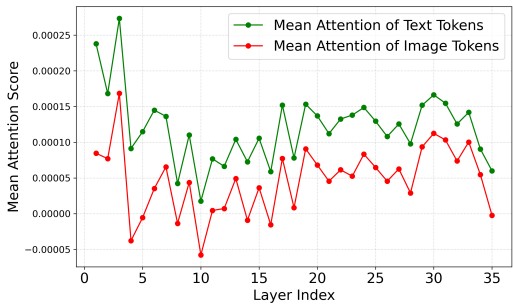

*Figure 2.* Layer-wise Mean Attention Scores (Text vs. Image). We report the average pre-softmax attention scores of the first generated token across all 35 Transformer layers on a ScienceQA subset. Text tokens consistently receive higher attention than visual tokens, indicating a systematic attention bias toward text.

language models tend to rely more on textual inputs when they conflict with visual evidence.

To rule out the possibility that the above phenomenon is merely an artifact of average-attention dilution caused by the large number of visual tokens, we further aggregate the total pre-softmax attention scores over textual and visual tokens, respectively. The results show that visual tokens still receive lower total pre-softmax attention scores than textual tokens, as reported in Appendix A.2. In addition, to verify that this phenomenon is not specific to the Qwen-VL family, we conduct further experiments on LLaVA-1.6-7B and observe a consistent attention bias toward textual tokens. More details are provided in Appendix A.3.

**Attention Dilution Caused by Long Textual Chains.** CoT-style generation typically produces long intermediate textual sequences (Zheng et al., 2023; Tan et al., 2024). As more text tokens are appended while the set of visual tokens remains fixed, the denominator in Eq. 9 increases, thereby diluting the relative attention mass assigned to visual tokens. This trend is consistent with recent findings (Chu et al., 2025; Sun et al., 2025; Jian et al., 2025; Liu et al., 2025a).

Consequently, $E_v$ is updated primarily by language-prior-dominated gradients, as visual tokens exert minimal influence on the generation process.

## 5. Methodology

### 5.1. Overview

We consider the standard multimodal reasoning setting: given an image $I$ and a question $q$, the goal is to produce a final answer $a_{\text{final}}$ grounded in $I$. As illustrated in Fig. 3, our framework consists of two modules: (1) a **Cognitive Reasoning Core (CRC)** instantiated as an LLM that conducts iterative reasoning and decides when additional visual

evidence is needed; and (2) a **Primary Visual Perception Module (PVP)** that returns *textualized visual evidence* from $I$ upon request. During inference, the CRC iteratively issues visual queries to the PVP when necessary, and integrates the returned information to produce $a_{\text{final}}$.

### 5.2. Cognitive Reasoning Core

The CRC serves as the central module of the framework. It is responsible for maintaining the global reasoning state and for determining whether the currently acquired evidence is sufficient to support a final decision, or whether additional perceptual evidence needs to be obtained.

**Design of the CRC.** We instantiate the CRC as an LLM rather than a VLM for three main reasons. First, from an extensibility perspective, a modality-agnostic reasoning core naturally generalizes to additional perceptual modalities (e.g., video or audio). Second, LLMs possess sufficient capacity to support such cognitive scheduling, as prior studies have shown their ability to actively identify missing information and seek additional evidence during multi-step reasoning under uncertainty (Hu et al., 2024). Finally, at comparable model scales, LLMs typically exhibit stronger abstract reasoning and logical control, whereas VLMs must balance additional cross-modal alignment objectives during training, often leading to trade-offs in pure reasoning capacity (Li et al., 2025; Zhou et al., 2025).

**Inputs and Reasoning State.** We define a *step* as one CRC invocation. At step $t$, the CRC takes as input a fixed instruction prompt $\text{Prompt}_{\text{CRC}}(q)$ (which contains the question $q$ and task specification) and a reasoning state $h_{t-1}$ that contains all previous interactions. In this work, we represent $h_{t-1}$ as the concatenation of (1) the cumulative reasoning trace $\{r_\tau\}_{\tau=1}^{t-1}$ produced by the CRC, and (2) the collection of textualized visual evidence $\{a_\tau^v\}_{\tau=1}^{t-1}$ returned by the PVP. At $t = 1$, $h_0$ is empty.

**Decision Mechanism.** At step $t$, the CRC produces an intermediate output

$$r_t = \text{LLM}(\text{Prompt}_{\text{CRC}}(q), h_{t-1}). \qquad (10)$$

The output $r_t$ expresses one of two intents: (i) issuing a new visual query for additional information, or (ii) producing the final answer to $q$. We introduce a routing function $g(\cdot)$ that deterministically parses $r_t$ into a structured decision:

$$o_t^c = \begin{cases} q_t^v, & \text{if } g(r_t) \in \mathcal{Y}_{\text{query}}, \\ a_{\text{final}}, & \text{if } g(r_t) \in \mathcal{Y}_{\text{answer}}, \end{cases} \qquad (11)$$

where $\mathcal{Y}_{\text{query}}$ and $\mathcal{Y}_{\text{answer}}$ are two disjoint sets of parsed outputs corresponding to a visual query and a final answer, respectively. Please note that the routing function is used

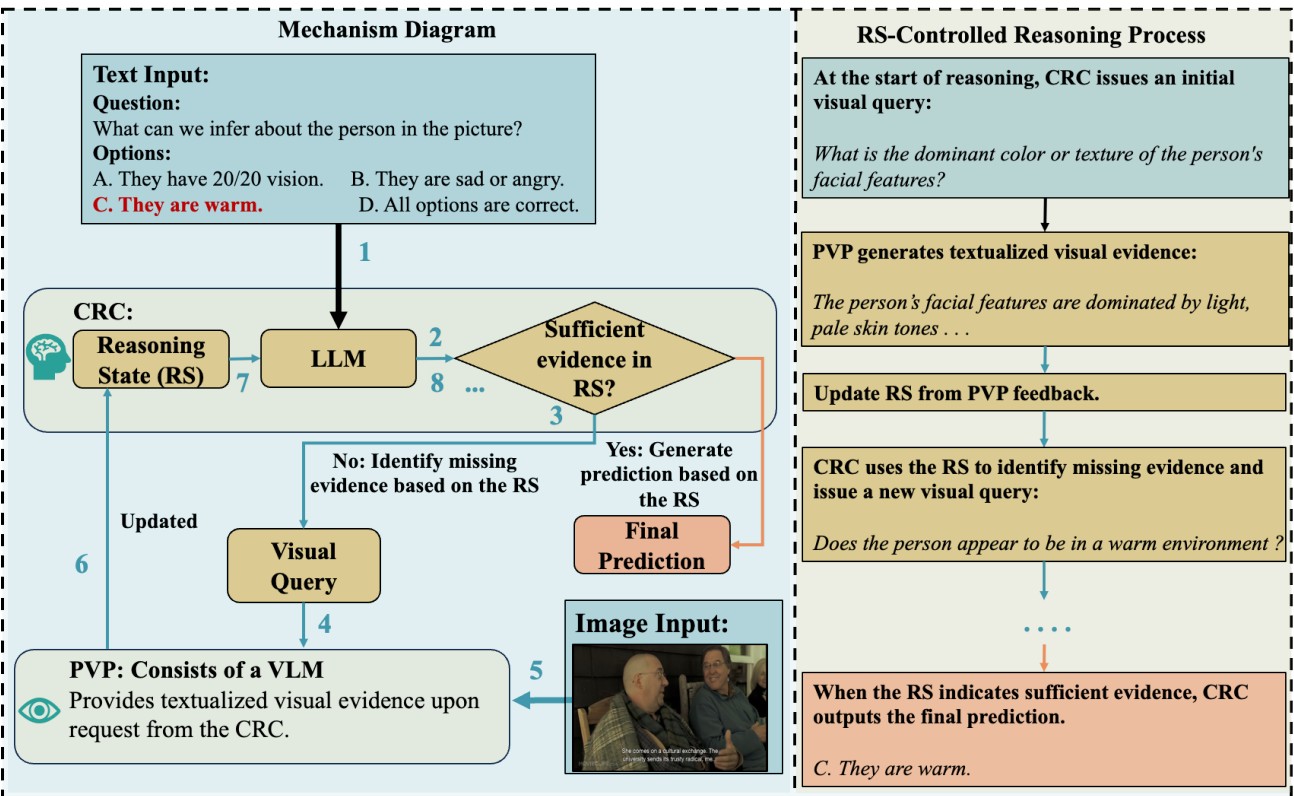

*Figure 3.* Overview of the CSMR architecture and its reasoning workflow. The left panel illustrates the overall structure of the CSMR, which consists of a CRC and a PVP. Given an input image and a question, the CRC maintains the current reasoning state and generates targeted visual queries to invoke the PVP when necessary. The PVP independently analyzes the original image and returns textualized visual evidence that answers the issued query. This evidence is then integrated into the CRC's reasoning state to support subsequent reasoning. The right panel presents a concrete example of reasoning. The CRC progressively generates visual queries based on the current reasoning state. Once the obtained textualized visual evidence is deemed sufficient, the CRC directly produces the final answer.

solely to parse the output of the CRC and does not influence the reasoning process of the CRC. Specifically, we constrain the CRC through prompting to follow the corresponding predefined formats, while $g(\cdot)$ simply applies regular expression matching to determine whether the output is a query $q_t^v$ or an answer $a_{\mathrm{final}}$.

**Reasoning state Update.** If $g(r_t) \in \mathcal{Y}_{\mathrm{answer}}$, the loop terminates and outputs $a_{\mathrm{final}}$. Otherwise, after receiving the PVP evidence $a_t^v$ , we update the reasoning state by

$$h_t = \mathrm{update}(h_{t-1}, r_t, a_t^v), \qquad (12)$$

where $\mathrm{update}(\cdot)$ aggregates the newly generated trace and visual evidence into the state. In this work, we simply implement $\mathrm{update}(\cdot)$ by concatenation, which preserves the full interaction history without introducing additional trainable parameters.

### 5.3. Primary Visual Perception

The PVP provides textualized visual evidence whenever the CRC issues a visual query. It takes as input the query $q_t^v$ and the image $I$, which are embedded into a prompting template $\mathrm{Prompt}_{\mathrm{PVP}}$ that instructs the model to answer $q_t^v$ based on the visual content in $I$. The PVP then invokes an (independent) VLM to produce:

$$a_t^v = \mathrm{VLM}(\mathrm{Prompt}_{\mathrm{PVP}}(I, q_t^v)). \qquad (13)$$

Textualized evidence is explicit and thus easier to verify than latent embeddings, which facilitates detecting visually inconsistent intermediate outputs.

### 5.4. Reasoning Process

Given $(I, q)$, the CRC starts from $h_0 = \emptyset$ and iterates Eq. 10. If $g(r_t) \in \mathcal{Y}_{\mathrm{query}}$, the CRC issues $q_t^v$ to the PVP, which returns $a_t^v$ via Eq. 13; the CRC then updates $h_t$ and proceeds to the next step. The loop terminates when $g(r_t) \in \mathcal{Y}_{\mathrm{answer}}$ or when the accumulated context reaches the maximum token budget $T_{\mathrm{max}}$.

*Table 1.* Main results on M3CoT, ScienceQA, and LLaVA-Bench In-the-Wild (LLaVA-W) with Qwen2 series backbones.

| Perception Backbone | Reasoning Backbone | Method | M3CoT ACC. | ScienceQA ACC. | LLaVA-W ROUGE-L |
|---|---|---|---|---|---|
| Qwen2-VL-7B | Qwen2-VL-7B | No-CoT | 43.6 | 56.3 | 32.7 |
| | | Multimodal CoT (Zhang et al., 2024b) | 40.1 | 51.3 | 30.7 |
| | | CCoT (Mitra et al., 2024) | 43.3 | 56.4 | 29.4 |
| | | SCAFFOLD (Lei et al., 2025) | 41.7 | 53.7 | 31.8 |
| | | ICoT (Gao et al., 2025) | 44.1 | 56.8 | 34.2 |
| Qwen2-VL-7B | Qwen2-7B | Caption | 40.9 | 67.7 | 29.1 |
| | | DDCoT (Zheng et al., 2023) | 39.0 | 71.9 | 26.4 |
| | | **CSMR (Ours)** | **45.7** | **78.2** | **34.3** |

*Table 2.* Ablation study on the M3CoT benchmark, analyzing the impact of different CRC control strategies on multimodal reasoning performance. All experiments are conducted under the zero-shot setting.

| Method | M3CoT Acc. | Time s/sample. |
|---|---|---|
| CSMR | **45.7** | 24.34 |
| *w/* Single-Query CRC | 40.0 | **12.35** |
| *w/* Pre-planned Visual Queries | 40.1 | 23.36 |
| *w/* Fixed-Step CRC | 42.7 | 72.48 |

# 6. Experiments

## 6.1. Datasets

We evaluate on three representative benchmarks covering multi-step reasoning, science-domain multimodal reasoning, and open-ended instruction following: M3CoT (Chen et al., 2024b), ScienceQA (Saikh et al., 2022), and LLaVA-Bench In-the-Wild (LLaVA-W) (Liu et al., 2024b). M3CoT emphasizes concise, visually grounded multi-step reasoning; ScienceQA focuses on cross-modal scientific problem solving across multiple subjects and grade levels; and LLaVA-W targets open-ended visual understanding with long-form answers annotated by GPT-4V.

## 6.2. Baselines

**No-CoT** directly generates answers from the given image and question without intermediate reasoning. **Caption** directly converts the image into a natural language description and conducts reasoning based on the generated caption. **CCoT** (Mitra et al., 2024) constructs a scene graph from the image to capture object-level and relational details, which is then used as structured input to guide reasoning. **DD-CoT** (Zheng et al., 2023) decomposes the original question into simpler sub-questions, solved by a VQA model when visual input is needed, and integrates the results for the final prediction. **SCAFFOLD** (Lei et al., 2025) adopts a non-parametric approach by introducing spatial anchors into

images that can be explicitly referenced in language, thereby enhancing the alignment and coordination between visual evidence and textual reasoning. **ICoT** (Gao et al., 2025) introduces interleaved visual–textual intermediate reasoning steps, enabling fine-grained multimodal reasoning by explicitly grounding textual rationales in image regions.

## 6.3. Implementation Details

Following the experimental settings of prior works (Gao et al., 2025; Li & Ma, 2025), we conduct experiments on the Qwen2 (Yang et al., 2024) series models. Specifically, Qwen2-VL-7B-Instruct is adopted as the perception backbone, while Qwen2-7B-Instruct serves as the reasoning backbone. For baseline methods, the results of No-CoT, CCoT, SCAFFOLD, and ICoT are directly taken from (Gao et al., 2025), where perception and reasoning are handled by a single unified model. Meanwhile, under a unified backbone setting, we implement DDCoT and Caption as comparison baselines. DDCoT is the most closely related to CSMR in terms of methodological paradigm, while the Caption method represents a simple vision-to-text baseline. The prompt template used by CSMR is provided in the Appendix B, while DDCoT adopts the prompt released in its original paper. For the Caption baseline, we employ a VLM to generate a textual description of the image and then answer the question based on the generated caption. All experiments are conducted under a zero-shot setting on two NVIDIA L40 GPUs. Additional implementation details and hyperparameter settings are reported in the Appendix C.

## 6.4. Main Results

Table 1 summarizes the performance of different methods on three multimodal reasoning benchmarks. As shown in the table, CSMR achieves the best performance across all benchmarks. Specifically, compared to ICoT, a strong representative of the unified multimodal representation paradigm, CSMR consistently attains superior results, suggesting that tightly coupled cross-modal fusion may not be a necessary condition for strong multimodal reasoning. Within the text-

*Table 3.* Results on M3CoT under different perception–reasoning backbone configurations.

| Perception Backbone | Reasoning Backbone | Method | M3CoT ACC. | Time s/sample |
|---|---|---|---|---|
| Qwen2-VL-7B | Qwen3-8B | Caption | 43.1 | **5.2** |
| | | DDCoT (Zheng et al., 2023) | 43.3 | 52.9 |
| | | **CSMR (Ours)** | **48.1** | 20.7 |
| Qwen3-VL-8B | Qwen3-8B | Caption | 45.2 | **6.8** |
| | | DDCoT (Zheng et al., 2023) | 50.3 | 54.6 |
| | | **CSMR (Ours)** | **53.6** | 23.1 |

centric pre-reasoning paradigm, CSMR also significantly outperforms DDCoT and Caption, demonstrating the advantage of cognitively scheduled reasoning over approaches that convert all visual information into text prior to reasoning. Notably, all performance gains are achieved without any additional training or fine-tuning, highlighting the structural benefits of our approach.

## 6.5. Ablation Study

To analyze the effect of different design choices, we conduct a series of ablation studies. First, removing dynamic visual querying by restricting the CRC to a single visual query results in a substantial drop in performance, with accuracy decreasing from 45.7% to 40.0%. This indicates that iterative, feedback-driven visual acquisition plays a critical role, and that a single-pass perception strategy is insufficient for the evaluated multi-step reasoning tasks. Second, we restrict the CRC to generate all visual queries for the entire reasoning process at the first step, instead of dynamically issuing queries during later reasoning steps. This constraint leads to a notable performance drop, with accuracy decreasing from 45.7% to 40.1%. This suggests that pre-planned visual queries, without intermediate reasoning feedback, often fail to align with the actual reasoning trajectory, highlighting the importance of dynamically acquiring visual information during step-by-step reasoning. Third, we fix the number of CRC–PVP interaction steps to seven, as the vast majority of samples require fewer than seven interaction steps in practice. Under this setting, performance further degrades, with accuracy dropping to 42.7%. This result indicates that flexible termination is essential for effective multimodal reasoning. It improves inference efficiency by avoiding unnecessary reasoning steps. More importantly, fixing the number of reasoning steps may introduce redundant information that interferes with the reasoning process and ultimately reduces accuracy. Overall, these ablation results demonstrate that the effectiveness of CSMR critically depends on the dynamic control mechanism of the CRC, rather than on static or pre-defined reasoning and querying strategies.



*Figure 4.* Comparison of hallucination rates between DDCoT and CSMR on M3CoT. Hallucinations are identified by GPT-5 based on inconsistencies between generated dialogues and image content. CSMR exhibits a lower hallucination rate than DDCoT.

## 6.6. Hallucination Analysis

This section investigates whether CSMR can effectively reduce hallucinations in multimodal reasoning. To ensure a fair comparison, we focus on DDCoT and CSMR, as both methods involve multiple invocations of the visual perception model during the reasoning process. We adopt M3CoT as the evaluation benchmark because its questions exhibit a strong reliance on visual evidence, thereby reducing the influence of linguistic priors. Specifically, we randomly sample 200 instances from M3CoT using the same random seed as in the main experiments. We then apply DDCoT and CSMR to this subset, respectively, while preserving their full multi-round reasoning dialogues. Finally, we employ GPT-5 as an automatic evaluator to assess each dialogue and identify whether it contains textual descriptions that are inconsistent with the visual evidence in the image, i.e., whether hallucinations occur. The experimental results are shown in Figure 4. As observed, compared to DDCoT, CSMR achieves a 9-percentage-point increase in the proportion of samples without hallucinations, indicating a clear advantage in hallucination control.

**Text Input:**
**Question:**
What is the primary business of the truck?
**Options:**
A. Transporting clothing.
**B. Selling food.**
C. Selling books.
D. Transporting cake.

**Image Input:**

**CSMR:**
**CRC:**
What is the primary activity of the truck in the image?
**PVP:**
The primary activity of the truck in the image is serving customers, as it appears to be a mobile bakery or dessert truck and there are people standing in front of it, possibly ordering food.
**CRC:**
Is the primary activity of the truck serving customers or transporting goods?
**PVP:**
The primary activity of the truck appears to be serving customers, as it has an open window and items inside, suggesting that it is selling goods, likely cakes.
**CRC:**
The Correct Option: B. Selling food

**DDCOT:**
**Sub-questions:**
(1) What is the type of goods the truck is carrying?
(2) What is the purpose of the truck's operation?
(3) What is the business associated with the goods being transported or sold?
**Sub-answers:**
(1) The truck is carrying goods related to baking, as indicated by the sign " Southern Cake Queen".
(2) The purpose of the truck's operation is to sell baked goods, specifically cakes, as indicated by the sign on the side of the truck.
(3) The business associated with the goods being transported or sold is "Southern Cake Queen".
**Rationale:**
Based on the description and the image provided, the truck is a mobile food truck with a sign that reads " Southern Bake Queen" or " Southern Cake Queen", indicating that it sells baked goods, specifically cakes.
**Output:**
D. Transporting cake.

*Figure 5.* Comparison of reasoning paths between DDCoT and CSMR. CSMR constructs a progressive, evidence-conditioned reasoning trajectory by dynamically generating sub-questions, while DDCoT relies on static and parallel sub-question decomposition, which leads to semantic drift and misaligned decision focus.

### 6.7. Case Study

To further illustrate the difference between the reasoning trajectories induced by CSMR and DDCoT, we analyze a concrete case study. As shown in Fig. 5, CSMR follows a dynamic, evidence-driven reasoning trajectory. Its reasoning process first acquires the necessary visual–semantic information by posing targeted questions, and then formulates more specific reasoning-oriented tasks, such as distinguishing between selling goods and transporting cargo. This questioning strategy forms a clear semantic progression, allowing each reasoning step to progressively narrow the decision space while continuously aligning with the core semantics of the original question, thereby effectively suppressing semantic drift during reasoning. In contrast, DDCoT induces a static reasoning trajectory by committing to a set of parallel questions before any concrete visual evidence is obtained. Lacking conditional dependencies, these questions tend to expand along the same semantic dimension. In this case, the reasoning process remains confined to identifying the type of goods carried by the truck, rather than shifting to the core discriminative dimension of selling versus transporting, which ultimately leads to semantic drift and hinders convergence to the correct conclusion.

### 6.8. Effectiveness Regimes of CSMR

This set of experiments examines how the relative reasoning capability between the perception backbone and the reasoning backbone influences the performance gains of

CSMR over DDCoT and the Caption baseline. We focus on DDCoT because it is the most closely related paradigm, while Caption serves as a simple vision-to-text reference. We consider two backbone configurations. In the first setting, Qwen2-VL-7B is paired with Qwen3-8B, where the reasoning backbone exhibits substantially stronger reasoning capability than the perception backbone. In the second setting, Qwen3-VL-8B is paired with Qwen3-8B; under image–text benchmarks, these two models can be regarded as providing comparable effective reasoning capability (Bai et al., 2025). As shown in Table 3, CSMR outperforms DDCoT by 4.8 accuracy points under the Qwen2-VL-7B and Qwen3-8B configuration. When switching to Qwen3-VL-8B and Qwen3-8B, this gain narrows to 3.3 points. Across both settings, the Caption baseline consistently yields substantially lower accuracy.

These results suggest that CSMR achieves the greatest performance gains when the reasoning backbone exhibits substantially stronger reasoning capability than the perception backbone. From a structural perspective, under this regime, CSMR centralizes global reasoning control in the reasoning backbone, allowing it to directly govern the acquisition and utilization of perceptual evidence. When the perception backbone already possesses strong autonomous reasoning ability, DDCoT can sufficiently leverage this capability, reducing the additional gains from centralized control. In terms of efficiency, although Caption remains the fastest due to its single-pass design, CSMR is consistently more efficient than DDCoT across both configurations. This effi-

ciency gain stems from the fact that in CSMR, the reasoning backbone actively decides when sufficient information has been gathered and halts further reasoning.

## 7. Conclusion and Future Work

We revisit a fundamental question in multimodal reasoning: how visual evidence should participate in reasoning. Existing paradigms either compress visual evidence before reasoning or weaken it through linguistic dominance in unified representations. We address this with a cognitive scheduling framework that decouples perception from reasoning and dynamically acquires visual evidence during reasoning. Experiments show consistent gains across benchmarks, suggesting that effective multimodal reasoning may depend more on principled evidence scheduling than tighter multimodal fusion. Additionally, the multi-model collaboration and dynamic visual querying in CSMR introduce additional inference overhead. Future work will explore techniques such as quantization to improve efficiency while maintaining reasoning performance (Jin et al., 2025; Ma et al., 2024a;b; 2025).

## Impact Statement

Recent advances in vision-language models (VLMs) and large language models (LLMs) have significantly increased the demand for stronger reasoning capability, better scalability, and lower training cost in multimodal reasoning tasks. This also raises two important concerns: (1) since the current implementation of CSMR is training-free, whether its performance ceiling may be limited in scenarios requiring extremely high accuracy or strong domain adaptation; and (2) compared with the single-pass inference of unified VLMs, whether the decoupled reasoning framework of CSMR may introduce higher inference overhead. In this work, we discuss both of these concerns.

For concern (1), we argue that the "training-free" property is not an intrinsic limitation of CSMR, but rather a design choice of the current implementation. Structurally, CSMR decouples the reasoning module (CRC) from the perception module (PVP), allowing them to be optimized either independently or jointly. In particular, the core capability of CRC is to actively identify missing information and dynamically acquire evidence based on the current reasoning state. Such information-seeking reasoning behavior can be further modeled and optimized through training. Meanwhile, PVP can also be fine-tuned to improve perception accuracy in specific domains. Therefore, CSMR does not fundamentally limit the achievable performance ceiling; instead, it provides a structured reasoning framework that is complementary to parameter learning.

For concern (2), we acknowledge that some unified VLM-based chain-of-thought (CoT) methods are more lightweight in terms of single-pass inference cost. However, the primary goal of CSMR is not to minimize inference cost, but to provide a decoupled capability scaling path. Experimental results show that overall performance can be consistently improved by strengthening the reasoning module while keeping the perception module unchanged. From this perspective, CSMR offers higher training efficiency: when stronger capability is required, only the LLM needs to be upgraded, without retraining the entire VLM, which is typically less costly than training a unified multimodal model of comparable scale. In addition, the decoupled structure also provides strong modality scalability. When introducing new modalities such as video or audio, CSMR only requires attaching the corresponding perception modules, whereas unified models often require joint modeling and retraining across all modalities, leading to substantially higher training costs.

## Acknowledgements

This work was supported by National Key R&D Program of China (No. 2023YFB4502804), the National Natural Science Foundation of China (No. U22B2051, No. U25B2066, No. 62302411).

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

*Table 4.* Accuracy on the selected ScienceQA subset (232 samples, grades 8–12) under different input configurations.

| Backbone | Question | Option | Image | Hint | Accuracy |
|---|---|---|---|---|---|
| Qwen3-VL | ✓ | ✓ | ✓ | ✓ | 92.67% |
| | ✓ | ✓ | ✓ | ✗ | 81.90% |
| | ✓ | ✓ | ✗ | ✓ | 68.10% |
| | ✓ | ✓ | ✗ | ✗ | 57.33% |

## A. Supplementary Experiment on ScienceQA

### A.1. Linguistic-Prior-Dominated Training

To validate our claim that a VLM can achieve a low training loss primarily by relying on linguistic priors, we conduct a controlled empirical study on a subset of the ScienceQA dataset. Specifically, we select 232 samples from grades 8–12 that include an image, question, option, and hint text. The hint text provides auxiliary textual information to help the model understand the question. Compared to lower-grade questions, these samples typically involve more complex reasoning and are more likely to require visual evidence, making them a more challenging testbed for language-only inference.

We adopt Qwen3-VL-8B, a recent large scale vision–language model, as the backbone to ensure that the observed behavior is not an artifact of limited model capacity. We evaluate four input configurations by selectively removing image and/or hint information. As shown in Table 4, even when no visual input is provided, the model achieves 57.33% accuracy without hints and 68.10% accuracy with hints, both substantially above random chance.

This observation suggests that, under current standard supervised training objectives, models may reduce training loss by relying heavily on linguistic priors, even in the absence of visual inputs. Consequently, during joint multimodal training, there is limited incentive for the visual encoder to maintain strict faithfulness to image evidence.

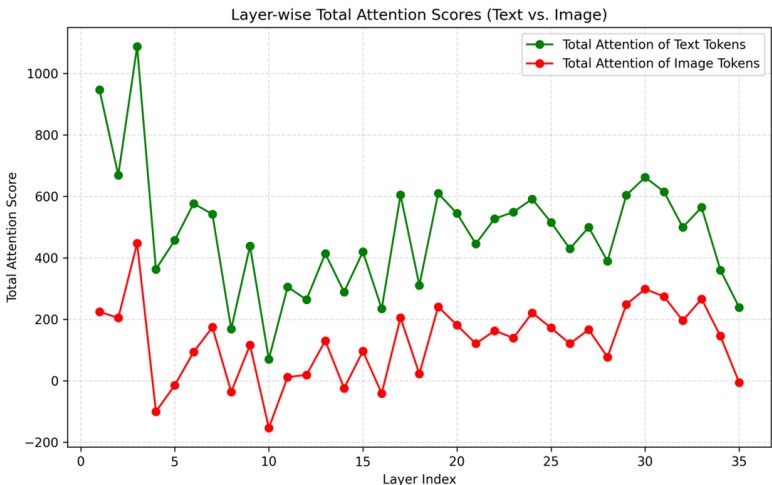

*Figure 6.* Layer-wise total pre-softmax attention scores allocated to text tokens and visual tokens during the generation of the first output token. Results are averaged over all attention heads and all samples in the ScienceQA subset using Qwen3-VL-8B. Across most layers, text tokens receive substantially higher attention mass than visual tokens, revealing a strong text-dominant attention bias during answer generation.

### A.2. Attention Distribution During Answer Generation

To further analyze the model's internal attention distribution during answer generation, we conduct an attention study on the ScienceQA subset selected in the above section, using the recent large-scale vision–language model Qwen3-VL-8B. Specifically, we focus on the attention allocation of the self-attention module over the input sequence when generating the first output token in the autoregressive decoding stage.

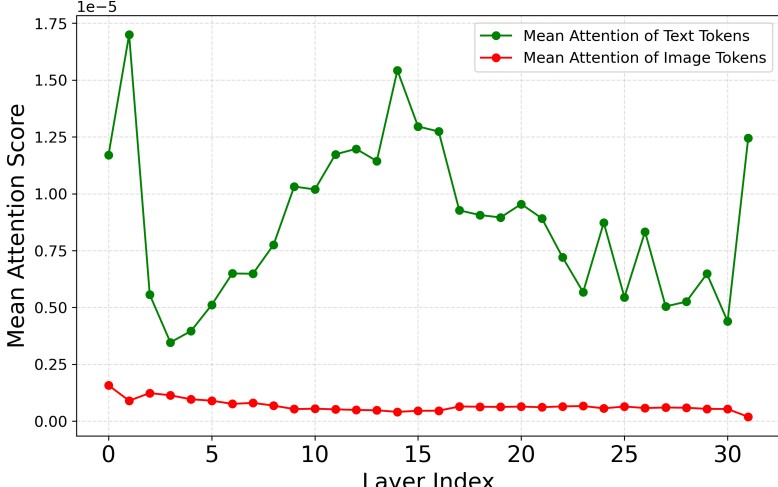

*Figure 7.* Layer-wise Mean Attention Scores (Text vs. Image). We report the average attention scores for the first generated token across all 32 Transformer layers on a subset of ScienceQA using LLaVA-1.6-7B. Text tokens consistently receive higher attention than visual tokens, indicating a systematic attention bias toward text.

For each sample, we concatenate the question, image, hint, and options as the model input, and extract the self-attention weights from every transformer layer. We then average the attention matrices across all attention heads within each layer to obtain a layer-level attention distribution; additionally, we average across the sample dimension to reduce variance due to individual example differences. Finally, we partition the input tokens by modality into visual tokens and text tokens, and compute the mean attention score for each group.

Meanwhile, we aggregate the total pre-softmax attention scores over text tokens and visual tokens at each transformer layer. Fig. 6 shows the layer-wise attention distribution averaged over all samples, where text tokens consistently receive higher total attention than visual tokens.

### A.3. Generalizability Verification of Attention Distribution Analysis

To examine whether the above attention analysis is affected by the visual token compression mechanism in the Qwen-VL family, we further conduct experiments on LLaVA-1.6-7B (Liu et al., 2024a). Since Qwen-VL models compress visual tokens and thus substantially reduce the number of visual tokens, the observed attention patterns may partly depend on this compression design. In other words, whether the same conclusion holds for models without visual token compression remains to be further verified. Unlike Qwen-VL models, LLaVA-1.6-7B does not adopt a comparable visual token compression mechanism, and therefore its input sequence contains a much larger number of visual tokens.

It is worth noting that the pre-softmax attention scores in LLaVA-1.6-7B are unnormalized logits and can take negative values. Therefore, directly summing the pre-softmax scores over tokens from different modalities does not admit a clear probabilistic interpretation. For this reason, we instead analyze the average attention score per token. Meanwhile, to investigate how the difference in token counts affects the overall attention allocation, we further report the total attention scores assigned to text tokens and visual tokens.

The experimental results are shown in Fig. 7 and Fig. 8. Specifically, Fig. 7 reports the average attention score per token, while Fig. 8 shows the total attention scores assigned to text and visual tokens. From the perspective of per-token attention, text tokens receive substantially higher attention than visual tokens, indicating that the model still tends to focus more on textual information at the unit-token level. From the perspective of total attention, although the number of visual tokens is much larger than that of text tokens, visual tokens dominate the total attention only in the early layers. As the layer depth increases, the total attention assigned to text tokens gradually surpasses that assigned to visual tokens. These results further suggest that, even in models without visual token compression, a clear text-oriented attention bias still emerges.

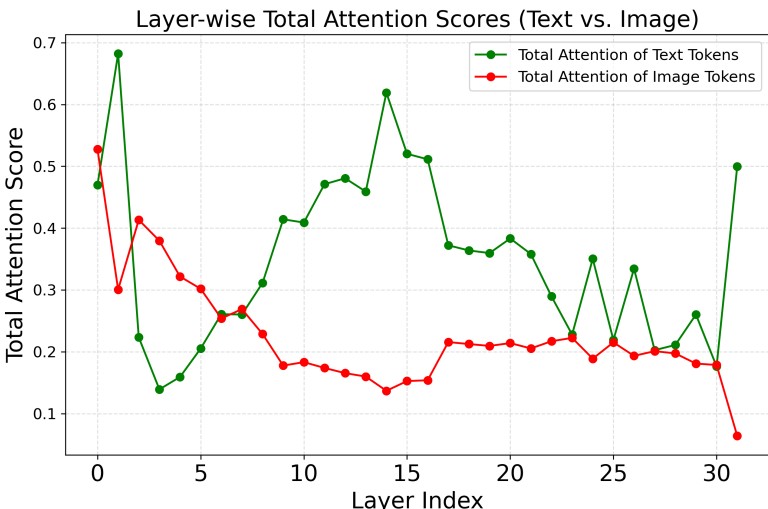

*Figure 8.* Layer-wise Sum Attention Scores (Text vs. Image). We report the total attention scores for the first generated token across all 32 Transformer layers on a subset of ScienceQA using LLaVA-1.6-7B. In most layers, text tokens receive higher attention scores than visual tokens, indicating a systematic attention bias toward text.

## B. Prompt Templates

The prompt template of the CRC is shown in Listing 1. The PVP answers the visual queries generated by the CRC by directly analyzing the image. Since the PVP does not involve decision-making or interaction control, we omit its prompt for brevity.

*Listing 1.* CRC Prompt Template

```
[Role and Input]
You are solving a multimodal reasoning task and are required to actively acquire visual
evidence from an image by asking visual questions.

You will receive:
- An input question related to the image.

[Phase 0: Integrated Problem Analysis]
In this phase, analyze the input question to identify the key visual evidence required for
 answering it.

[Phase 1: Visual Questioning]
In this phase, ask visual questions about the image to obtain information necessary for
answering the input question.
Each question should be grounded in the image content and aim to reduce uncertainty in the
 reasoning process.
After receiving the corresponding visual evidence, incorporate it into your latest
analysis.

[Phase 2: Answer Decision]
Based on the updated analysis, determine whether the available visual evidence is
sufficient to answer the input question with confidence.
- If not, return to Phase 1 and ask another visual question.
- If so, output the final prediction to the input question.
```

## C. Hyperparameter Settings

*Table 5.* Hyperparameter settings for reasoning and perception backbones.

| Hyperparameter | Reasoning Backbone | Perception Backbone |
|---|---|---|
| Temperature | 0.3 | 0.7 |
| Top-$p$ | 0.9 | 0.9 |
| Top-$k$ | 30 | 80 |
| Max tokens | 2048 | 512 |
| $T_{max}$ | 6000 | 6000 |
| Repetition penalty | 1.0 | 1.0 |
| Max model length | 8192 | 8192 |

