# OpenReview forum: "Look on Demand: A Cognitive Scheduling Framework for Visual Evidence Acquisition in Multimodal Reasoning"
_ICML.cc/2026/Conference — ICML 2026 regular_

### Official Review · Reviewer_34TD · 2026-02-22

**Soundness:** 3
**Presentation:** 3
**Significance:** 3
**Originality:** 3
**Overall Recommendation:** 5
**Confidence:** 5

**Summary:**

This paper focuses on a central question in multimodal reasoning: when and how visual evidence should be incorporated into the reasoning process. The authors observe that existing approaches mainly fall into two categories. One converts images into text in a static manner before reasoning, which inevitably leads to the loss of fine-grained visual information. The other performs end-to-end reasoning within a unified vision–language representation space, where language priors often dominate and weaken the influence of visual evidence. The limitations of both paradigms stem fundamentally from suboptimal designs regarding the timing and manner in which visual evidence is introduced.

To address this issue, the paper proposes the CSMR framework. In this framework, a language model serves as a cognitive reasoning core that explicitly maintains the reasoning state and dynamically decides whether additional visual information is required. An independent visual perception module is introduced to analyze the raw image on demand and return targeted visual evidence. This design enables multi-round, demand-driven visual queries and allows the reasoning process to terminate early once sufficient evidence has been obtained.

The paper further analyzes the causes of visual representation distortion in unified multimodal models, arguing that language-dominated training objectives and self-attention mechanisms systematically suppress the influence of visual tokens. This analysis provides empirical motivation for the structural design choices underlying CSMR.

Experimental results on benchmarks such as M3CoT, ScienceQA, and LLaVA-W show that, under zero-shot settings, CSMR improves both accuracy and reasoning efficiency. Ablation studies and case analyses demonstrate that dynamic visual querying and reasoning-state-aware scheduling play a critical role in reducing hallucinations and enhancing consistency with visual evidence.

Overall, the paper proposes and validates a new multimodal reasoning paradigm centered on cognitive scheduling. The results suggest that improvements in multimodal reasoning do not necessarily require tighter representation-level fusion, but can instead be achieved through structurally grounded, on-demand incorporation of visual evidence.

**Compliance With Llm Reviewing Policy:**

Affirmed.

**Final Justification:**

See my response to the authors, my major concerns were all addressed, so I am comfortable recommending the paper.

**Key Questions For Authors:**

**Question 1: On the Generality of the “Language-Dominant Attention Leading to Visual Unfaithfulness” Analysis**

The analysis of how language-dominant attention may undermine the faithfulness of visual representations in unified VLMs is insightful. However, the supporting empirical evidence is primarily based on Qwen-3VL. Could the authors clarify whether this phenomenon is expected to generalize to other mainstream VLMs, and if so, on what grounds?



**Question 2: Structural Benefits of CSMR and Reasoning Token Overhead**

The authors attribute the performance gains of CSMR to its structural design, particularly the decoupling of cognitive scheduling and perception–reasoning, and also report certain efficiency advantages in inference time. In this context, could the authors further clarify whether CSMR introduces additional reasoning tokens compared to baseline methods?



**Question 3: Relationship Between CSMR and Tool-Calling–Based Methods**

CSMR bears some formal resemblance to tool-calling–based approaches, such as ReAct, in that both involve language-model-driven iterative interactions. Could the authors more explicitly articulate the key structural differences between CSMR and such methods, and clarify in what aspects CSMR provides irreducible advantages over existing tool-calling paradigms?



**Question 4: Potential Limitations of the Train-Free Design**

As a train-free approach, CSMR offers clear advantages in terms of generality and ease of deployment. Do the authors anticipate that, in scenarios requiring extremely high precision or strong domain adaptation, the absence of any training may limit the achievable performance ceiling? Is there potential to combine the CSMR framework with lightweight training or adaptation mechanisms?

**Limitations:**

**Constructive suggestions for improvement:**

1. **Provide a clearer discussion of limitations**: For example, clarify CSMR’s reliance on the capability gap between the reasoning backbone and the perception module; discuss its robustness when the perception model is relatively weak or when the visual evidence itself is noisy; and address the computational overhead and latency risks introduced by multi-round visual querying.
2. **Discuss potential negative societal impacts**: Although the method reduces hallucination to some extent, in safety-critical scenarios (e.g., healthcare, autonomous driving, and legal decision support), incorrect or insufficient scheduling of visual evidence may still lead to misleading conclusions. In addition, repeated invocation of perception models may amplify data bias or introduce privacy risks.
3. **Outline possible mitigation directions**: For instance, incorporating uncertainty estimation, human-in-the-loop verification, or application-specific constraints in high-risk settings to avoid over-reliance on automated reasoning outcomes.

**Strengths And Weaknesses:**

### 1.Technical Soundness

**Strengths:**
 The paper is technically sound overall. The authors provide a clear mechanistic analysis of the issue that language priors dominate and dilute visual evidence in unified multimodal representation paradigms, supported by reasonable empirical evidence through attention statistics and controlled experiments. The proposed CSMR framework is structurally concise, does not rely on additional training, and is evaluated with well-designed experiments. The ablation studies effectively validate the necessity of dynamic visual querying and flexible termination mechanisms.

**Weaknesses:**
 The decision-making process of the CRC relies entirely on the language model’s generation and parsing behavior. Its stability and failure modes—such as premature termination or excessive querying—have not been systematically analyzed, which may affect reliability in more complex or noisy scenarios.

### 2. Presentation

**Strengths:**
 The paper is well structured, with a clear and coherent logical flow from problem motivation to mechanism analysis, method design, and experimental validation. The authors clearly distinguish between the two dominant multimodal reasoning paradigms in the introduction and methodology, and use illustrative diagrams to intuitively highlight the key differences in information flow between CSMR and existing approaches. The roles of the CRC and PVP are well defined, the reasoning process is described with sufficient detail for reproducibility, and the prompt templates and hyperparameter settings provided in the appendix further enhance practicality.

**Weaknesses:**
 Some analytical discussions overlap with experimental results and could be further streamlined to improve conciseness. In addition, the boundary between CSMR and certain recent interactive or tool-augmented methods could be articulated more directly.



### 3. Significance

**Strengths:**
 The paper addresses an important and timely question: how and when visual evidence should be incorporated into multimodal reasoning. This issue is particularly critical given the current trend toward unified VLMs. The work offers a valuable reflection on the implicit assumption that tighter modality fusion necessarily leads to stronger reasoning, and demonstrates an alternative path through structural design. From a practical perspective, CSMR achieves consistent performance gains without additional training, making it attractive for resource-constrained or rapid-deployment scenarios. Moreover, the analysis suggesting that centralized cognitive control becomes more effective when the reasoning backbone is substantially stronger than the perception backbone provides useful empirical guidance for future model composition and system design.

**Weaknesses:**
 At present, the impact of the work is primarily confined to multi-step multimodal reasoning tasks. Its applicability to a broader range of vision–language tasks, such as purely perceptual or generation-oriented settings, remains unclear.



### 4. Originality

**Strengths:**
 The originality of the paper lies mainly in its perspective and structural design. Rather than introducing entirely new model components, the authors systematically incorporate the concept of *cognitive scheduling* into the multimodal reasoning process, reorganizing the interaction between perception and reasoning. By explicitly positioning the LLM as a cognitive control core and treating the visual module as an on-demand external evidence source, the framework offers a new explanatory lens for understanding and mitigating hallucination and semantic drift in multimodal reasoning. Additionally, the paper’s systematic analysis of structural attention biases in unified representation paradigms constitutes a meaningful complementary insight to existing methods.

**Weaknesses:**
 From a methodological standpoint, CSMR shares high-level conceptual continuity with existing tool-calling or decomposition-based reasoning approaches (e.g., DDCoT). Its novelty lies more in the clarity of mechanism organization and motivation rather than in proposing an entirely new algorithmic paradigm. As such, the contribution is better characterized as a recombination and deepening of understanding rather than a fundamentally disruptive innovation.



### 5. Overall Assessment

Overall, the paper is technically solid, with a clear motivation and a strong alignment between method design and experimental results. Its main contributions are a systematic analysis of structural issues in multimodal reasoning and a reasonable and effective cognitive-scheduling-based solution. While there remain aspects that could be further refined or extended, the work presents a clear and well-substantiated contribution centered on mechanistic understanding and structural improvement.

---

> ### Author Rebuttal · Authors · 2026-03-30
>
> Thanks for your kind review and useful suggestions! Please let us know if the following answers your questions and addresses your concerns, we are very happy to provide additional information if needed.
>
>
> > - (Question 1) Analysis of the Generality of Language-Dominant Attention Leading to Visual Unfaithfulness.
>
> To show that our analysis is not based on an isolated case, we further reproduced the experiment in Section 4.2 of the paper on LLaVA-v1.6-Vicuna-7B. This model contains 32 Transformer decoder layers. We selected layers 5, 10, 15, 20, 25, and 30, and examined the distribution of attention scores over text tokens and image tokens. The results are shown in the table. As can be seen, the trend is consistent with our observations on Qwen3-VL-8B: across these layers, the attention scores of text tokens are generally higher than those of image tokens.
>
> Please note that these are pre-softmax attention logits, and thus the values are negative; accordingly, values closer to zero indicate higher similarity.
>
> | Layer Index | 5 | 10 | 15 | 20 | 25 | 30 |
> |-------------|---|----|----|----|----|----|
> | Text Tokens | -0.026 | -0.021 | -0.019 | -0.022 | -0.027 | -0.030 |
> | Image Tokens | -0.047 | -0.041 | -0.040 | 0.043 | -0.044 | -0.046 |
>
>
> > - (Question 2) Comparison of Token Overhead Between CSMR and Baseline Methods
>
> From a methodological perspective, DDCOT is one of the baselines most closely related to our approach. Therefore, under the same experimental setting as in Section 6.6 of the paper, we measured the average token overhead per sample for both CSMR and DDCOT, including both input and output tokens.
>
> The results show that the average token overhead of CSMR is 1214.2, whereas that of DDCOT is 2031.6. These results indicate that CSMR not only achieves lower reasoning time than DDCOT, but is also more efficient in terms of token overhead.
>
>
> > - (Question 3) Relationship Between CSMR and Tool-Calling–Based Methods
>
> Tool-calling methods such as ReAct perform reasoning by imitating the “thought–action–observation” trajectories provided in demonstrations, and are therefore essentially trajectory imitation-driven. In contrast, our method introduces an explicit reasoning state and makes decisions at each step based on the current reasoning state, making it fundamentally reasoning-state-driven. This difference leads to two key distinctions.
>
> First, in terms of the purpose of queries, queries in ReAct often serve to continue the behavioral pattern demonstrated in the examples and may not strictly correspond to the actual missing information required by the current task. By contrast, in our method, each query is driven by the current reasoning state and is specifically aimed at acquiring the critical information missing at each stage, making it more tightly aligned with the task.
>
> Second, regarding the termination mechanism, when the demonstrations are relatively long, the model may be biased toward continuing to issue queries, even if sufficient information has already been obtained. In contrast, our method can explicitly determine, based on the reasoning state, whether the available information is sufficient, and can therefore terminate the reasoning process proactively, improving both efficiency and stability.
>
>
> >-  (Question 4) Potential Limitations of the Train-Free Design
>
> We believe that “train-free” is not an intrinsic limitation of CSMR, but rather its current implementation form. When higher precision or stronger domain adaptation is required, CSMR can likewise be combined with training to further improve performance.
>
> Structurally, CSMR decouples the reasoning module (CRC) from the perception module (PVP), allowing them to be optimized either independently or jointly. In particular, the core capability of CRC is to actively identify missing information and dynamically acquire the required evidence based on the current reasoning state under uncertainty. This information-acquisition-driven reasoning capability can be explicitly modeled and further trained.
>
> Meanwhile, the PVP can also be fine-tuned to improve perceptual accuracy in specific domains. Therefore, CSMR does not impose a limit on the achievable performance ceiling; rather, it provides a structural reasoning framework that is complementary to parameter learning.
>
>
>
> >-  (Weakness 1) Generalizability of the Method
>
> By decoupling reasoning from perception, our method is structurally better suited for cross-modal generalization. Unlike models that rely on unified multimodal representations, CSMR does not require retraining a large-scale model that jointly covers multiple modalities. Instead, it can be extended simply by introducing the corresponding perception module for a new modality, such as video or audio. This modular design not only improves flexibility in generalization, but also avoids the substantial training cost associated with joint multimodal modeling.

---

> > ### Author Rebuttal · Reviewer_34TD · 2026-04-01
> >
> > I appreciate the authors’ rebuttal. It addressed my concerns clearly and helped me better understand what the paper is really about. More importantly, I now see this work as a genuinely different direction, not just another incremental variant. What I find most valuable is the shift in perspective it brings. Rather than continuing to push within the dominant unified vision–language reasoning paradigm, this work reframes the problem at a higher level and introduces a more flexible way to organize multimodal reasoning. In my view, this kind of reframing is more important than incremental improvements, because it opens up new design space and provides a clearer path for the field to evolve. Overall, I think this paper can have a meaningful positive impact on the community, beyond the specific results. I strongly support acceptance and will raise my score to 5.

---

> > > ### Author Response · Authors · 2026-04-01
> > >
> > > Thank you very much for your encouraging feedback and constructive suggestions. They have been very helpful in improving the manuscript.

---

### Official Review · Reviewer_dYy5 · 2026-02-26

**Soundness:** 2
**Presentation:** 3
**Significance:** 2
**Originality:** 2
**Overall Recommendation:** 4
**Confidence:** 3

**Summary:**

This paper introduces an innovative framework designed to dynamically integrate visual evidence via multi-step LLM reasoning. A key strength of this approach is the use of short-phrase queries during evidence retrieval, which successfully mitigates the "token dilution" issue often found in end-to-end models. By decoupling reasoning from perception, the proposed framework utilizes visual signals more robustly than integrated VLMs, ensuring that the visual context is not overshadowed by the linguistic bias of the text tokens.

**Compliance With Llm Reviewing Policy:**

Affirmed.

**Final Justification:**

The article provides an easy to upgrade framework, and the author's supplementary content solves my concern. Although this model may not have priority in reasoning efficiency, it also provides an extensible idea.

**Key Questions For Authors:**

1) How is the routing function $g(\cdot)$ implemented and what are the specific criteria used to determine whether a response constitutes the final answer?
2) Although Section 5.2 explains the rationale for using an LLM instead of a VLM, the experiments in Appendix A1 demonstrate that incorporating images can improve accuracy, which suggests that using a VLM in CRC might yield more refined reasoning results. Is there any experimental evidence to support or clarify this?
3) According to the official M3CoT leaderboard, the zero-shot CoT performance of Qwen2-VL is 57.8, which is higher than the results reported in this paper. What is the reason for this discrepancy?
4) Are there any comparisons with VLM-based CoT methods using a single-step reasoning approach to demonstrate the effectiveness and reasoning cost advantages of the proposed method?
5) The adoption of a combined LLM and VLM framework essentially doubles the total number of parameters. Can the authors provide a comparison with a standalone VLM of a comparable parameter size to ensure a fair evaluation of the model's efficiency and performance?

**Limitations:**

Can this approach of using an LLM to pose questions and a VLM to provide answers handle more complex problems? While this method can reference options for multiple-choice questions when formulating queries, the questions generated for non-multiple-choice tasks without direct access to the image might be irrelevant or biased, potentially leading to reasoning errors.

**Strengths And Weaknesses:**

Strengths:
The primary objective of this paper is to address the critical issue of visual information being lost or diluted during multimodal reasoning. The logical progression of the manuscript is rigorous as the authors first provide empirical evidence to confirm the existence of this problem before clearly articulating the principles of CRC and PVP and their synergistic mechanisms. Furthermore, the experimental results demonstrate that both modules contribute significantly to the overall performance and that the proposed approach consistently outperforms existing state-of-the-art methods.

Weaknesses:
The proposed framework faces several critical issues regarding technical clarity and experimental validation. First, the implementation details of the routing function $g(\cdot)$ and the termination criteria for the reasoning process are insufficiently described, making it difficult to assess the system's reliability. Second, while the paper advocates for a decoupled LLM-based reasoning approach, the authors' own supplemental data suggests that integrating visual information earlier might yield superior results. Finally, the framework appears heavily reliant on the presence of multiple-choice options to guide query generation, which potentially limits its robustness and generalizability when addressing complex, open-ended tasks where the reasoning agent lacks direct visual grounding.

---

> ### Author Rebuttal · Authors · 2026-03-31
>
> Thank you very much for your thorough and highly constructive review. We have revised the paper accordingly based on your suggestions. We hope the following clarifications address your questions and concerns. We would be happy to provide further details if needed.
> > - (Question 1) Implementation of the routing function and termination criterion.
>
> In our method, the routing function g(⋅) is used solely to parse the output of the CRC and does not influence the reasoning process of the CRC. Specifically, the CRC produces only two types of outputs: query or answer. To this end, we constrain the CRC through prompting to follow the corresponding predefined formats, while g(⋅) simply applies regular expression matching to determine whether the output is a query or an answer.
>
> If the output does not match either format, we only prompt the model to regenerate according to the specified format. Therefore, once g(⋅) identifies that the CRC output matches the answer format, the reasoning process terminates and the final answer is returned.
>
> > - (Question 2) Why is the CRC implemented with an LLM rather than a VLM?
>
> The key reason for using an LLM rather than a VLM as the CRC is to change how language priors operate.
>
> In unified VLMs, textual and visual tokens compete for attention in a shared representation space, so language priors directly affect attention allocation and can suppress visual participation.
>
> In contrast, when the CRC is an LLM, it does not take visual input. Thus, language priors no longer act through text–vision attention competition, but instead help identify missing visual information and guide subsequent queries. Their role therefore shifts from suppressing visual participation to guiding visual acquisition. If the VLM were used as the CRC, the model would directly access all visual information at the initial stage, causing language priors to once again act through the allocation of attention between visual and textual information. Consistent with this, replacing the CRC with Qwen2-VL-7B yields 45.2% accuracy on M3CoT, below the 45.7% achieved by Qwen2-7B.
>
> > - (Question 3) On the official M3CoT leaderboard, the zero-shot CoT performance of Qwen2-VL is higher than the result reported in this paper.
>
> We would like to clarify that the reported accuracy of 57.8% on the official M3CoT leaderboard comes from Wang et al. [1], whose result is based on additional training of Qwen2-VL. In contrast, among the methods on the leaderboard that, like ours, use Qwen2-VL without fine-tuning, the highest reported zero-shot accuracy is 40.2%.
> > - (Questions 4 and 5) When compared with single VLM-based CoT baselines, is the additional overhead in parameter scale and inference time of the proposed method sufficiently justified by its performance gain?
>
> We agree that some VLM-based CoT methods are lighter in per-instance inference cost. However, the core contribution of CSMR is not to reduce inference cost, but to provide a decoupled capability scaling path.
>
> As shown in Table 1 and Table 3, performance can be improved by strengthening only the reasoning backbone while keeping the perception module unchanged. From this perspective, CSMR offers better training efficiency: if further capability improvement is desired, one only needs to upgrade the LLM, which is typically less costly than training a VLM of comparable scale [2].
>
> Moreover, this decoupled structure is also easier to extend to additional modalities. When introducing a new modality, CSMR only requires plugging in the corresponding perception module, whereas unified models typically require joint modeling and training across all modalities, which often incurs substantially higher training cost.
> > - (Weakness 1) Whether CSMR depends on multiple-choice options for query generation, and therefore may have limited generalization to open-ended complex tasks that lack direct visual grounding?
>
> Thank you for raising this important concern. We agree that multiple-choice options can indeed provide additional constraints for query generation, and that in open-ended tasks, if the model lacks an initial understanding of the image content, it may generate queries with limited relevance.
>
> However, we would like to emphasize that options are only one form of available task context, rather than a prerequisite for query generation. For open-ended tasks, the CRC can first establish an initial understanding of the image through coarse-grained queries about the overall scene, major objects, or global layout, and then progressively issue more fine-grained queries relevant to the task, thereby reducing query drift in a coarse-to-fine manner. This mechanism is also supported by our experimental results: on the non-multiple-choice dataset LLaVA-W, our method likewise outperforms all baseline methods.
>
> [1] Enhancing the Reasoning Ability of Multimodal Large Language Models via Mixed Preference Optimization. (Wang et al., 2024)
>
> [2] Qwen2.5-VL Technical Report. (Bai et al., 2025)

---

> > ### Author Rebuttal · Reviewer_dYy5 · 2026-04-01
> >
> > The author answered my questions and emphasized the purpose of the article, and I fully understood the article. So I raised the score to 4.

---

> > > ### Author Response · Authors · 2026-04-01
> > >
> > > Thank you very much for your positive feedback. Your constructive comments have helped us improve the clarity and overall quality of the manuscript.

---

### Official Review · Reviewer_rQAw · 2026-03-09

**Soundness:** 2
**Presentation:** 3
**Significance:** 2
**Originality:** 2
**Overall Recommendation:** 4
**Confidence:** 4

**Summary:**

The authors propose CSMR, a cognitive scheduling framework for visual evidence acquisition in multimodal reasoning. It employs a large language model as the core reasoning engine and dynamically invokes a vision–language model to acquire visual evidence during the reasoning process. CSMR achieves state-of-the-art (SOTA) performance on three datasets.

**Compliance With Llm Reviewing Policy:**

Affirmed.

**Final Justification:**

The authors have fully addressed my concerns, and I have accordingly increased my score twice. The authors are encouraged to incorporate the key points and revisions addressed in the rebuttal into the revised manuscript. The proposed technical approach shows promise, and additional experimental analyses have been provided during the rebuttal stage.

**Key Questions For Authors:**

1. In Figure 2, what are the average numbers of visual tokens and text tokens in the evaluation data? Is it possible that the larger number of visual tokens leads to diluted average attention scores for visual tokens? What are the total attention weights assigned to visual tokens and text tokens, respectively?

**Limitations:**

The experimental results should be validated on a broader range of models to strengthen the conclusions.

**Strengths And Weaknesses:**

Strengths:

1. The authors propose the CSMR framework to mitigate the bias introduced by directly using visual features extracted by VLMs and the information loss caused by generating captions as intermediate representations.

2. The paper is clearly written and well presented.

3. The proposed method achieves state-of-the-art performance on three datasets.


Weaknesses:
1. The framework obtains visual evidence by asking visual questions to the VLM. Since the generation of these visual questions are related to the LLM, the approach may still be affected by textual priors.

2. Model behavior with and without image inputs can differ significantly. Using only 232 samples from ScienceQA without image inputs to argue that the visual encoder in jointly trained VLMs shows limited faithfulness to visual evidence is not sufficiently convincing. When images are absent, the model may simply guess the answer based on other textual cues in the question. Therefore, the results under the no-image setting may not fully reflect the model’s faithfulness to visual information when images are actually provided. A more rigorous analysis would require evaluating models trained with different datasets and training objectives.

3. Similar conclusions have already been discussed in prior work [1].

4. The experimental results should be validated on a broader range of models to strengthen the conclusions.

[1]Words or Vision: Do Vision-Language Models Have Blind Faith in Text? CVPR2025

---

> ### Author Rebuttal · Authors · 2026-03-31
>
> Thank you very much for your valuable comments. We have added an additional experiment and revised the paper accordingly based on your suggestions. We hope the following clarifications address your concerns. If needed, we would be happy to provide further details.
>
> > - (Question 1) The lower average attention assigned to image tokens than to text tokens may not directly indicate modality preference, but could instead result from an averaging dilution effect caused by a substantially larger number of image tokens.
>
> In fact, the number of visual tokens is smaller than that of text tokens. According to our statistics, after preprocessing with the Qwen3-VL processor, the average number of visual tokens is 99.6, while the average number of text tokens is 144.4.
>
> In addition, the total attention allocated to visual and text tokens is reported in Appendix Figure 6. Following your suggestion, we have revised the paper and moved this figure to the main text. Based on these results, we believe that the lower average attention received by visual tokens cannot be attributed to their larger quantity.
>
> > - (Weakness 1) Since the visual queries are generated by the LLM, the proposed method may still be affected by textual priors.
>
> We agree that our method is also influenced by textual priors. However, its mechanism is fundamentally different from that of unified VLMs.
>
> In unified representation spaces, text and visual tokens compete for shared attention, and textual priors tend to bias attention toward text tokens, thereby systematically suppressing the contribution of visual information.
>
> In contrast, in our framework, textual priors do not act on visual representations or attention allocation; instead, they are used to identify missing visual information in the current reasoning state and to guide subsequent evidence acquisition. Results in Section 6.6 provide evidence that our method effectively reduces the probability of hallucinations during reasoning.
>
> > - (Weakness 2) Using only 232 samples from ScienceQA without image input is insufficient to support the claim that visual encoders in VLMs have limited faithfulness to visual evidence.
>
> Following your suggestion, we expanded this analysis to all ScienceQA samples that include images. The results show that the model achieves 91.03% accuracy with full input, while still reaching 66.39% accuracy when images are removed.
>
> We would like to clarify that this experiment is not intended to prove that current training objectives necessarily cause visual encoders to deviate from image-grounded evidence. Rather, its purpose is to show that existing VLMs can still substantially reduce training loss even when relying only on textual priors. This suggests that the visual encoder may play a relatively limited role in loss optimization.
>
> > - (Weakness 3) Similar conclusions have already been discussed in prior work.
>
> Thank you for pointing out this relevant line of work. We have added the corresponding discussion and citation in the revised paper. We agree that the prior study also reveals a preference toward the textual modality. However, our work differs in two important respects.
>
> a) Mechanism: The prior work mainly identifies and validates the existence of this phenomenon, whereas our work further analyzes its underlying causes from the perspectives of training objectives and internal attention dynamics.
> b) Method: The prior work focuses on diagnosing modality preference, while we propose a new reasoning framework that structurally changes how visual evidence is acquired and integrated.
>
> Following your suggestion, we also conducted a broader survey of related literature and identified two additional highly relevant and insightful works [1,2], which have been incorporated into the related work section.
>
> > - (Weakness 4) The experimental results should be validated on a broader range of models to strengthen the conclusions.
>
> We have additionally included experiments on LLaVA-1.6-7B. Please refer to Question 1 in Reviewer 34TD for details.
>
> [1] Seeing is Believing: Mitigating Hallucination in Large Vision-language Models via CLIP-Guided Decoding. (Den et al., 2024)
>
> [2] Energy-Guided Decoding for Object Hallucination Mitigation. (Liu et al., 2025)

---

> > ### Author Rebuttal · Reviewer_rQAw · 2026-04-02
> >
> > The authors have fully addressed my concerns, and I have accordingly increased my score.

---

> > > ### Author Response · Authors · 2026-04-03
> > >
> > > Thank you very much for your positive feedback and constructive comments.
> > > > - (Question 1) What is the computational cost of the proposed method?
> > >
> > > We compare our method with two categories of approaches: text-centric pre-reasoning methods and unified multimodal representation methods.
> > >
> > > **Comparison with text-centric pre-reasoning methods.** We use DDCOT as a representative baseline. Such methods decompose the task into multiple subtasks before inference and execute them sequentially. As a result, even when sufficient information has already been obtained at an intermediate stage, the model must still complete all subtasks, leading to redundant computation. In contrast, our method dynamically decides whether further queries are needed based on the current reasoning state, and can terminate early once sufficient evidence has been collected.
> > >
> > > We verify this advantage in three aspects. 1) Token cost. Under the same setting as Section 6.6, the average token cost per sample is 1214.2 for CSMR and 2031.6 for DDCOT, showing that our method is more token-efficient. 2) Inference time. As shown in Table 3 of our paper, with the same backbone model, DDCOT takes about 2.4× longer than our method on a single sample. 3) Parameter and training cost. Since our method uses the same backbone model as DDCOT, both methods have the same parameter count and backbone training cost.
> > >
> > > **Comparison with unified multimodal representation methods.** From the perspective of inference cost, we agree that such methods are generally lighter. However, our method provides a decoupled capability-upgrading path. As shown in Tables 1 and 3 of our paper, when the perception module is fixed, improving only the reasoning backbone already yields better performance. This suggests better training efficiency: to further improve the system, one only needs to upgrade the LLM, which is typically less costly than training a VLM of comparable scale [1]. Moreover, the decoupled design is naturally more extensible to new modalities: adding a new modality only requires introducing the corresponding perception module, whereas unified models usually require joint modeling and training across all modalities, often at substantially higher training cost.
> > >
> > > > - (Question 2) How well does the method generalize to other tasks? In particular, do the observed conclusions still hold when the number of visual tokens increases?
> > >
> > > To examine whether the text-biased attention phenomenon identified in our paper also exists in models that encode images into a large number of visual tokens, we conduct an additional analysis on LLaVA-1.6-7B, a representative model of this type.
> > >
> > > **Token statistics:** After preprocessing with the LLaVA-Next-7B Processor, the average numbers of image and text tokens are:
> > >
> > > |Image Tokens|Text Tokens|
> > > |---|---|
> > > |1439.3|181.1|
> > >
> > > As shown, the number of image tokens is substantially larger than that of text tokens.
> > >
> > > **Comparison of average attention:** Our experiments show that the average attention per text token remains higher than that per image token. Detailed results can be found in Reviewer 34TD (Question 1).
> > >
> > > **Layer-wise attention distribution analysis:** To rule out the possibility that the lower average attention on image tokens is merely due to dilution by their larger quantity, we further analyze the attention distribution across all 32 Transformer layers. Since the attention logits in LLaVA-1.6-7B are negative, directly summing them is not meaningful; we therefore analyze the total post-softmax attention distribution instead. The results are shown below:
> > >
> > > |Layer|1|5|10|15|20|25|30|32|
> > > |---|---|---|---|---|---|---|---|---|
> > > |Text|0.47|0.16|0.41|0.62|0.36|0.35|0.26|0.50|
> > > |Image|0.53|0.32|0.18|0.14|0.21|0.19|0.18|0.06|
> > >
> > > From the table, we observe that although image tokens receive relatively higher attention in a few individual layers, text tokens dominate in most layers.
> > >
> > > **Overall attention statistics:** Averaging across all layers, image tokens receive 0.22 total attention, whereas text tokens receive 0.35, indicating that text tokens receive about 1.55× more attention. Since mainstream CoT-based methods continuously generate additional text tokens during reasoning, this text-dominant bias may further intensify over time.
> > >
> > > **Conclusion and generalization:** These results show that the text-biased attention phenomenon identified in our paper is not limited to the Qwen series, but also exists in models with a large number of visual tokens, such as LLaVA. More importantly, our method aims to restructure how visual information participates in multimodal reasoning, rather than relying on a specific model architecture or task format. This makes the framework broadly applicable across models and task settings. For discussion of its applicability to high-precision training scenarios and more open-ended complex tasks, please refer to Reviewer 34TD (Question 4) and Reviewer dYy5 (Weakness 1).
> > >
> > > [1] Qwen2.5-VL Technical Report (Bai et al., 2025).

---

### Decision · Program_Chairs · 2026-04-30

**Decision:**

Accept (regular)

**Comment:**

The paper proposes CSMR, a framework that decouples reasoning and perception by leveraging an LLM for multi-step reasoning and dynamically invoking a VLM for visual evidence acquisition, effectively mitigating token dilution and achieving a clear improvement.

Several questions were raised during the review phase, including routing design, textual prior bias, visual faithfulness, experimental validation, and fairness of model comparisons. The reviewers noted that the authors’ rebuttal has fully addressed these concerns. The authors also discussed the impact of the training-free design. I would like to highlight that the authors should more clearly articulate the broader impact of this work in a dedicated impact statement section, including the provided discussion of the proposed training-free strategy. Conditioned on this minor revision, overall, this is a solid work. I recommend acceptance.